# Deformation Texture of Bulk Cementite Investigated by Neutron Diffraction

**DOI:** 10.3390/ma15134485

**Published:** 2022-06-25

**Authors:** Nozomu Adachi, Haruki Ueno, Satoshi Morooka, Pingguang Xu, Yoshikazu Todaka

**Affiliations:** 1Department of Mechanical Engineering, Toyohashi University of Technology, 1-1 Hibarigaoka, Tempaku, Toyohashi 441-8580, Aichi, Japan; ueno@martens.me.tut.ac.jp (H.U.); todaka@me.tut.ac.jp (Y.T.); 2Materials Sciences Research Center, Japan Atomic Energy Agency, 2-4 Shirakata, Tokai, Naka 319-1195, Ibaraki, Japan; morooka.satoshi@jaea.go.jp (S.M.); xu.pingguang@jaea.go.jp (P.X.)

**Keywords:** cementite, deformation texture, neutron diffraction

## Abstract

Understanding the deformation mechanism of cementite such as on a slip plane is important with regard to revealing and improving the mechanical property of steels. However, the deformation behavior of cementite has not been well investigated because of the difficulty of sample preparation given the single phase structure of cementite. In this study, by fabricating bulk single phase cementite samples using the method developed by the authors, the deformation texture formed by uniaxial compression was investigated using both electron back scatter diffraction and neutron diffraction. The fabricated sample had a random texture before the compression. After applying a compressive strain of 0.5 at 833 K, (010) fiber texture was formed along the compressive axis. It has been suggested from this trend that the primary slip plane of cementite is (010).

## 1. Introduction

Pearlite is one of the common phases found in steel which has a lamellar structure with ferrite and cementite. It is well-known that pearlitic steels have superior strength-ductility balance. Pearilitic steels exhibit a high work-hardening rate, and wire-drawn pearlitic steels reach tensile strengths of 6.3 GPa [1]. Pearlitic steels are therefore used as industrial material such as suspension cable for suspension bridges and tire reinforcement steel wire. Although the primary reason of the high work-hardening rate of pearlitic material is thought to be a decrease of lamellar spacing by wire-drawing [2], the origin of such extremely high strength is not yet fully understood since wire-drawing causes the partial chemical decomposition of cementite and the amorphization of cementite. Recent in situ neutron diffraction experiments under tensile loading has revealed that stress partitioning between ferrite and cementite plays an important role in the high strength of pearlitic steels [3]. Since a stress partitioning is generated by a difference of mechanical properties between phases such as strength, deformability, and plastic anisotropy, understanding the deformation mechanism in both phases are quite important to revealing the origin of the strength of pearlite. Whereas the deformation mechanism of ferrite is well-established, the literature reporting the deformation mechanism of cementite is limited. Cementite has an orthorhombic unit cell with a space group of *Pnma* (No. 62) which contains 12 Fe atoms and 4 C atoms [4]. While one type of bonding between Fe atoms is metallic bonding, that between Fe and C atoms is characteristic of covalent bonding, which should affect the slip deformation mechanism of cementite. Recent in situ synchrotron X-ray diffraction experiments have revealed the lattice constants of cementite to be *a* = 0.5084 nm, *b* = 0.6747 nm, and *c* = 0.4525 nm at room temperature [5]. The time-of-flight neutron powder diffraction experiment also reported similar lattice constants [6]. Inoue et al. has investigated the deformation behavior of cementite in cold-rolled carbon steels by observing the dislocation structure using transmission electron microscopy. They have reported that slip planes (100), (010), and (100) were observed and a slip direction of [100] on (010) and [010] on (100) was suggested [7]. However, since cementite is generally formed as one of constitute phases in a multiphase microstructure, the obtainable size of the cementite sample is limited in micrometer scale. Therefore, the deformation behavior of cementite has not been systematically investigated.

Our research group has previously proposed the method to fabricate a bulk single phase cementite sample using the mechanical ball milling and pulse current sintering (PCS) processes [8,9]. This method enabled us to investigate the properties of cementite alone such as hardness [10], elastic properties [11], and the hydrogen permeation property [12]. In this study, by employing the method, we investigated the deformation behavior of cementite by means of a neutron diffraction experiment.

## 2. Materials and Methods

Pure Fe (particle diameter *d*_p_ < 150 μm) and graphite (*d*_p_ < 20 μm) powders with 99.9% purity were mixed to be stoichiometric composition of cementite (i.e., Fe-25at%C) and subjected to mechanical ball milling (MM) using the conventional horizontal ball mill. The conditions of MM performed in this study are summarized in Table 1.

After the MM process, supersaturated Fe + C solid solution is formed. The MM processed powder was subjected to the PCS (Sumitomo Coal Mining Co., Ltd., Tokyo, Japan) at 1173 K for 900 s at a compressive pressure of 50 MPa in a vacuum and then furnace cooled. Cementite is formed during the PCS process, and bulk cementite with a diameter of 10 mm and a height of 8–10 mm was consequently obtained in this study. By using this method, a bulk sample having nearly 95 vol% of cementite can be fabricated [12].

In order to investigate the deformation behavior of cementite, a compressive strain of 0.5 was applied to the bulk cementite sample by compression test using the PCS. The compression tests were performed at 833 K in a vacuum.

The bulk cementite samples before and after the compression test were mechanically polished and finished using a colloidal silica, and then the microstructure was observed using a Schottkey field emission scanning electron microscope (SEM, SU5000 Hitach High-Tech Corporation, Schaumburg, IL, USA) attached with electron back scatter diffraction (EBSD, TexSEM Laboratories, Inc., Draper, UT, USA). EBSD profiles were analyzed using the OIM analysis software.

The as-sintered sample and the compressed sample were subjected to a neutron diffraction (ND) experiment to investigate the deformation texture of cementite. The ND experiment was performed using the diffractometer for residual stress analysis (RESA) at the Japan Research Reactor No.3 (JRR-3) of the Japan Atomic Energy Agency (JAEA). The sample for the ND experiment has a diameter of 10 mm and a height of 10 mm. As the compressed sample has a thickness of 2 mm, five samples were stacked to make a total thickness of 10 mm. In the ND experiment, seven diffraction peaks (002), (201), (211), (102), (112), (221), and (122) were measured while rotating a sample along χ and ϕ with a step of 5° and 15° (see Figure 1 for the definition of χ and ϕ). Here, CD and RD means compressive direction and radial direction, respectively. The obtained ND profiles were analyzed by using MAUD (Materials Analysis Using Diffraction, version 2.97) software to calculate pole figures [13].

## 3. Results and Discussion

### 3.1. Texture of Bulk Cementite before Compression Test

Figure 2 shows the phase, inverse pole figure (IPF), and kernel average misorientation (KAM) maps of the as-sintered bulk cementite sample obtained by the EBSD. The phase map indicates that the as-sintered cementite sample has a volume of 96% cementite and contains 4% ferrite, which is in good agreement with our previous investigation using neutron diffraction [12]. We have also investigated the porosity of the sintered sample through the bulk density measurement based on Archimedes’ method [12]. It found that the as-sintered sample had a porosity volume of 3.2 %. The cementite phase in the sample has an equiaxed grain shape with an average grain size of 0.6 μm. The KAM map shows that each grain in the sample has relatively low misorientation angle, suggesting that the dislocations introduced by the MM was recovered during the sintering.

{001}, {010}, and {100} pole figures were also obtained from the EBSD to determine the texture in the as-sintered sample. The results are shown in Figure 3. The maximum texture intensity was only 1.4, showing that the as-sintered sample has a random texture.

As an EBSD method can collect information only from a surface of a sample, the internal texture of the sample can differ from the texture obtained by an EBSD. Considering this problem, the ND experiments were performed in this study. A neutron beam has very high penetration depth against steels over 10 mm, which enables the measuring diffraction from the internal microstructure. Figure 4 shows the examples of the selected seven diffractions of cementite obtained by the ND experiments at χ of 90°.

It can be seen from the profiles that there is no clear difference in peak intensities with varying ϕ, suggesting that the sample has random texture. From the result of peak fitting of each diffraction profile, lattice constants of *a* = 0.5085 nm, *b* = 0.6748 nm, and *c* = 0.4521 nm were obtained, which is a quite reasonable value considering previous studies [4,5,10]. By using diffraction profiles measured along wide range of χ and ϕ, pole figures were calculated. The obtained pole figures are shown in Figure 5. The pole figure shows a maximum texture intensity of 1.3 and indicates that the sample has random texture, which is a result consistent with the result of EBSD, as shown in Figure 3.

### 3.2. Evolution of Deformation Texture in Bulk Cementite by Compression

Figure 6 illustrates the IPF maps observing the CD-RD and RD-RD planes. It can be seen that grains are slightly elongated along RD in the CD-RD plane and are equiaxed in RD-RD planes. By randomly selecting 100 cementite grains from each image, aspect ratios of the cementite grain are calculated to be 0.9 and 1.7 in the RD-RD and CD-RD planes, respectively. This trend is quite reasonable considering that cementite grains were plastically deformed by the introduction of a compressive strain of 0.5. The KAM map is also shown in Figure 6e. It seems that the misorientation in each grain in the compressed sample is smaller compared with that in the as-sintered sample, showing that the atmospheric temperature of 833 K was high enough to cause the dynamic recovery of dislocations introduced by the compression.

It should be noted that a proportion of grains having an orientation close to (010) is large in the RD-RD plane (Figure 6d), suggesting that the deformation texture was formed. Figure 7 shows the corresponding pole figures. The axisymmetric pole figure can be clearly seen, in other words, a (010) fiber texture was formed along CD. As shown in Figure 8, this trend was also confirmed by the analysis based on the ND profiles performed with the same way as shown in Figure 4 and Figure 5.

In the case of uniaxial compressive deformation, it is generally known that primary slip plain gradually aligns perpendicular to the compression axis (i.e., CD) with increasing compressive strain. The (010) fiber texture found in this study therefore indicates that the primary slip plain of cementite is (010) at least under an elevated temperature of 833 K. As was mentioned in the introduction, Inoue et al. has reported that the possible slip planes of cementite are (100), (010) and (001) [7]. By investigating macroscopic deformation texture by employing the ND experiment, it has been concluded that the primary slip plane is (010). One of the possible reasons why (010) being primary slip plane is chemical bondings across (010). As was mentioned in the introduction section, cementite crystal consists of both metallic and covalent bondings. Figure 9 shows the schematic illustration of the atomic structure and the type of bonding between atoms in the cementite crystal. The trace of (010) was also shown in the figure with a blue line. It can be seen that (010) is the only plane where bondings across the plane consist of only metallic bonding. Since the strength of the metal bonding is much lower compared with covalent bonding, the critical resolved shear stress required to activate slip deformation is thought to be lower compared with other planes. The results obtained in this study showed that the method we used is effective to investigate primary slip plane. However, as shown in von-Mises’ criterion [14], at least five independent slip systems are required to accommodate deformation to an arbitrary shape. A detailed investigation is therefore essential to understand the more detailed deformation mechanism of cementite.

## 4. Conclusions

This study investigated the deformation texture of cementite by preparing nearly 100 vol% of cementite in bulk shape using the mechanical ball milling and pulse current sintering processes. The texture of the samples was measured by both the electron back scattering pattern and the neutron diffraction experiments. The as-sintered bulk cementite sample showed random texture with equiaxed grains having a grain size of 0.6 μm. After the application of a compressive strain of 0.5 at 833 K, the cementite grains were plastically deformed and showed an elongated shape along the radial direction of the sample. The (010) of cementite aligned perpendicularly to the compression direction, showing that the primally slip plane of cementite is (010).

## Figures and Tables

**Figure 1 materials-15-04485-f001:**
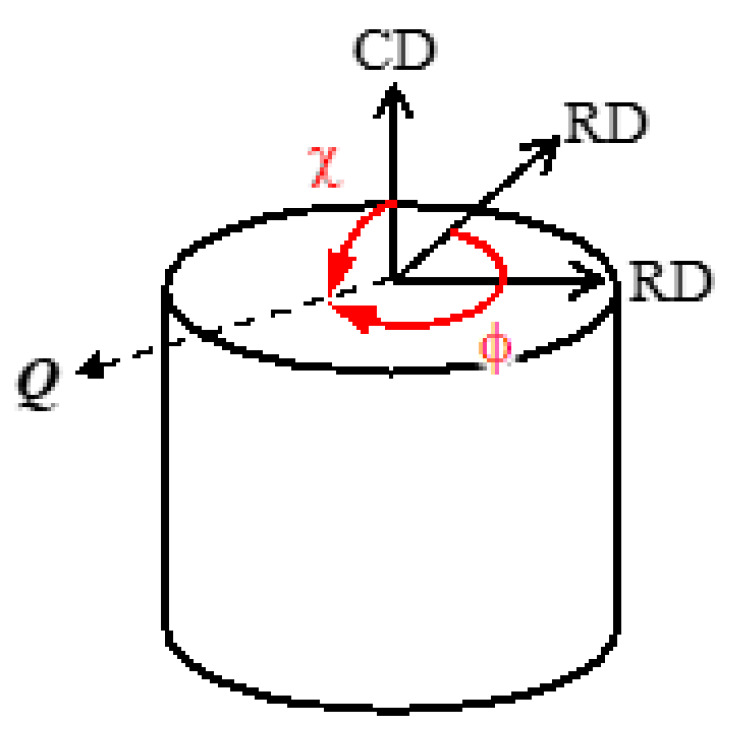
Schematic illustration showing the definition of χ and ϕ in the ND experiments, where *Q* is the neutron scattering vector.

**Figure 2 materials-15-04485-f002:**
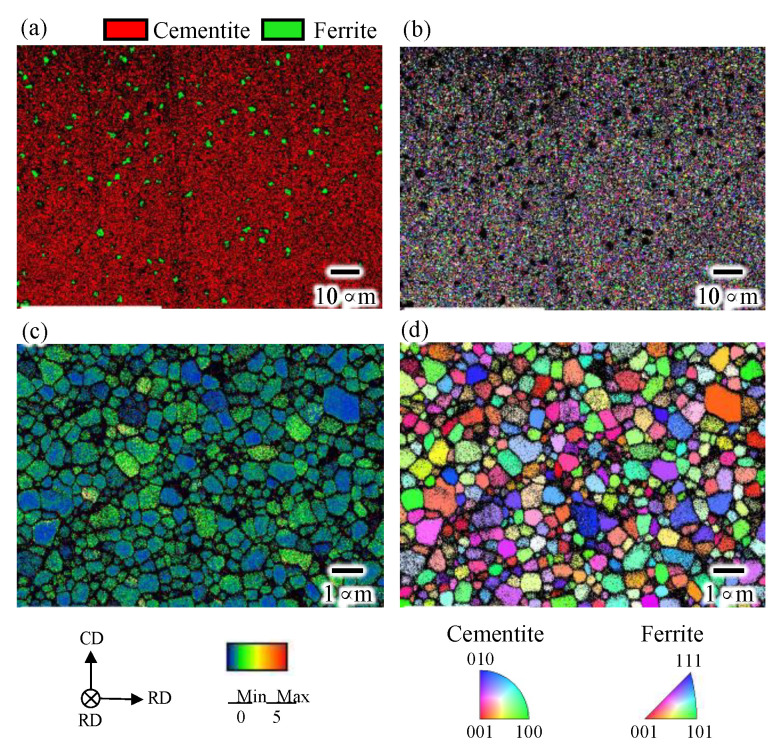
(**a**) Phase, (**b**,**d**) inverse pole figure (IPF), and (**c**) kernel average misorientation maps of the as-sintered bulk cementite sample.

**Figure 3 materials-15-04485-f003:**
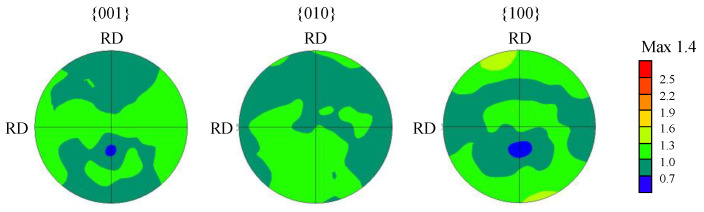
{001}, {010}, and {100} pole figures of as-sintered samples obtained by the EBSD.

**Figure 4 materials-15-04485-f004:**
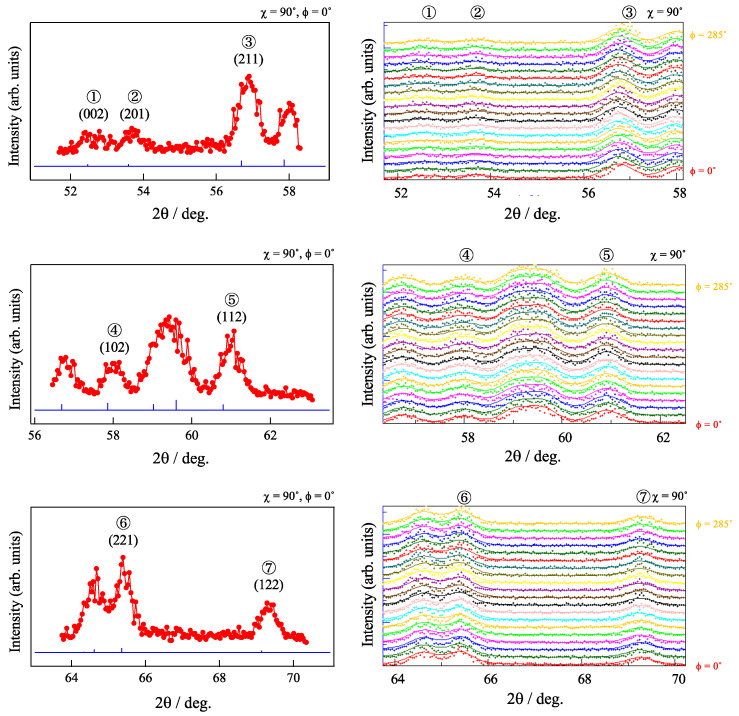
Typical neutron diffraction patterns obtained in the as-sintered sample. The blue lines in the left portion of the figure shows the expected peak positions of cementite. Each solid line in the right portion of the figure is the result of peak fitting.

**Figure 5 materials-15-04485-f005:**
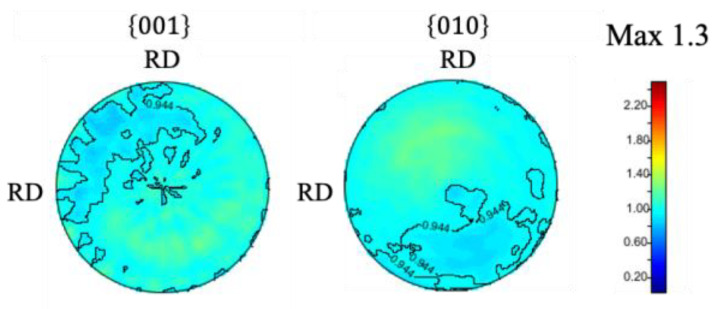
Pole figure maps of as-sintered sample obtained from neutron diffraction profiles.

**Figure 6 materials-15-04485-f006:**
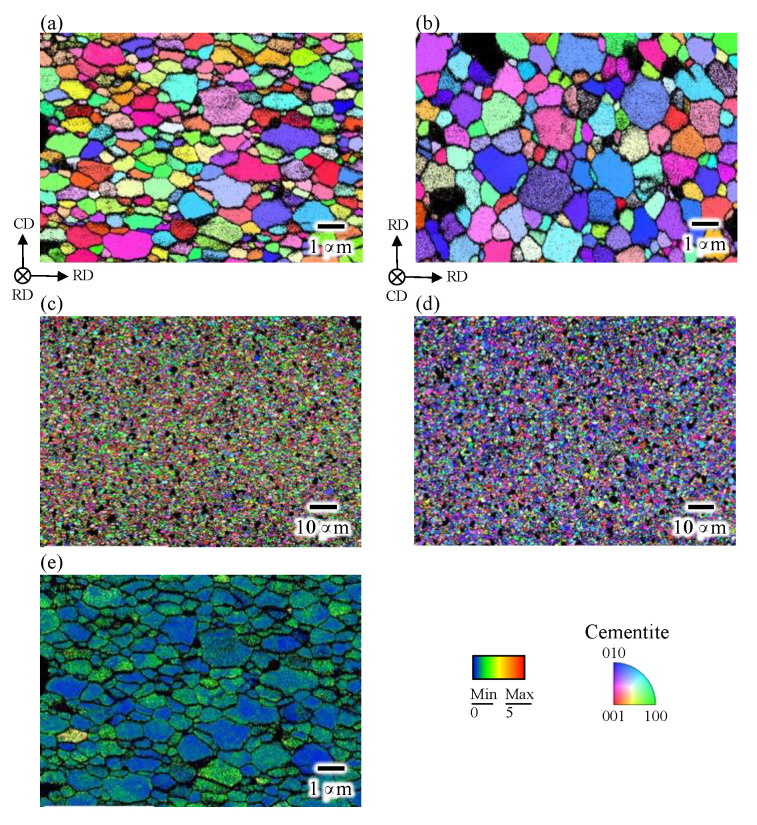
IPF maps of the compressed sample in (**a**,**c**) CD-RD and (**b**,**d**) RD-RD plains. (**e**) KAM maps.

**Figure 7 materials-15-04485-f007:**
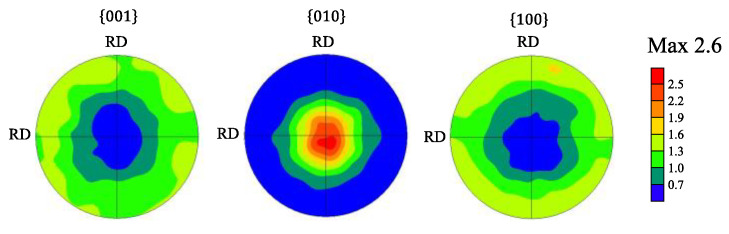
Pole figures of the compressed sample obtained by the EBSD.

**Figure 8 materials-15-04485-f008:**
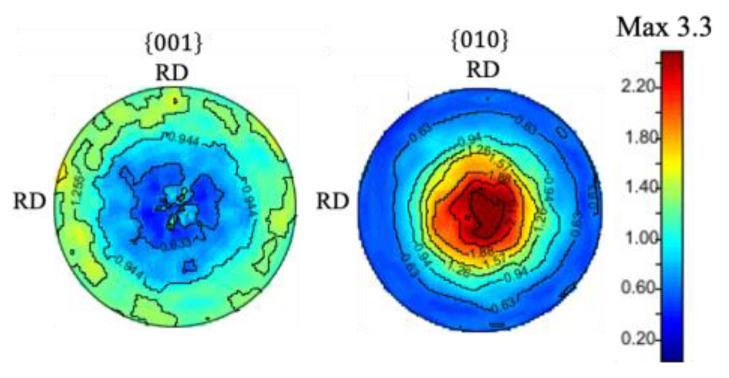
Pole figures of the compressed sample obtained by the neutron diffraction experiments.

**Figure 9 materials-15-04485-f009:**
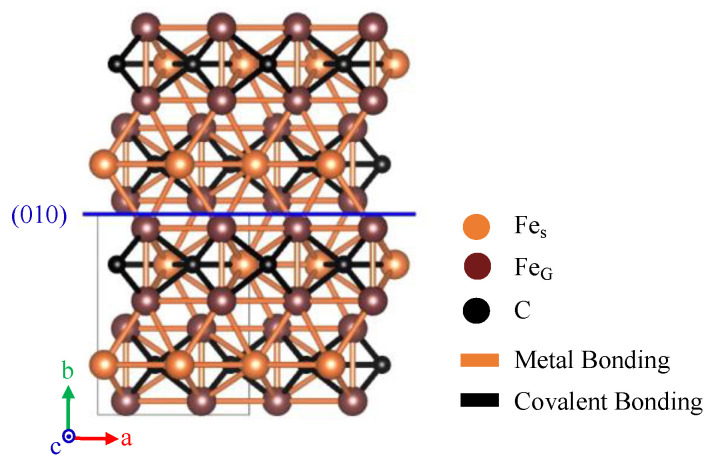
Schematic illustration of cementite observed from [001], where black and orange lines show that a type of bonding between atoms are covalent and metallic bonding, respectively.

**Table 1 materials-15-04485-t001:** Conditions of mechanical ball milling (MM) performed in this study.

Milling Receptacle	JIS SUS304 Stainless Steel (Inner Diameter ϕ128 mm, Volume 1.7 L)
Milling media	JIS SUJ2 Bearing Steel ball (Diameter ϕ25 mm)
Milling media weight	3800 g
Powder weight	38 g
Milling time	360 ks
Rotation speed	95 rpm
Atmosphere	Ar

## Data Availability

The data presented in this study are available on request from the corresponding author.

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
