# Peer review of "Deformation Texture of Bulk Cementite Investigated by Neutron Diffraction"

_materials, 2022, doi:10.3390/ma15134485_

Round 1

Reviewer 1 Report

Review

 Cementite plays a key role in formation of strength and plasticity of structural steels. In this regard, experimental ascertainment of the regularities of its plastic deformation is of key importance for the development of models of plastic deformation and fracture of steels.

The specific feature of this work lies in study of the compression texture on macro-sized specimens. In addition, when determining the texture parameters, two methods were used - back scatter diffraction and neutron diffraction.

The manuscript describes in detail the technique for obtaining macro-sized cementite specimens, their plastic deformation by compression. The results of crystallographic texture study are presented. Utilisation of mutually complementary methods made it possible to  substantiate the type of texture.

In general, this article is written in the tradition of good materials science works.

Remarks

1. The article doesn’t provide information on the porosity of macro-sized specimens, which is inherent in sintered materials. Apparently, the presence of porosity determined the high brittleness of cementite samples, as a result of which such high (833 K) temperature was used for plastic deformation by compression.

2. In Fig.9, instead of a schematic illustration, it is advisable to present the crystal structure of cementite with an indication of the lattice parameters. This will increase the information content of the article.

3. Authors should write a short conclusion.

 Summary

Taking into account above remarks, the manuscript may be accepted for publication.

Author Response

Dear Reviewer,

Thank you very much for your useful comments. I have revised the manuscript after the comments. Our response to your comments are as follows.

  • 1. The article doesn’t provide information on the porosity of macro-sized specimens, which is inherent in sintered materials. Apparently, the presence of porosity determined the high brittleness of cementite samples, as a result of which such high (833 K) temperature was used for plastic deformation by compression.
    • Thank you for your valuable comment. I have added the information about porosity of as-sintered sample. Of course, as the reviewer suggested, pores in a sintered material can cause embrittlement. However, influence of porosity on brittleness is uncertain, because cementite is known as brittle material at around room temperature even if there is no pores, which is common fracture behavior in carbides. Therefore, we chose compression test temperature of 833 K to induce slip deformation. Unfortunately, we currently do not have the information about morphology of fracture surface after compression test because we have stopped the test before fracture. Therefore, it is difficult to conclude that the pores in our sample determines brittleness of our sample. From above reasons, I decided not to mention about the relation between porosity and deformation. I hope the revised version of the manuscript is acceptable for the reviewer.
  • 2. In Fig.9, instead of a schematic illustration, it is advisable to present the crystal structure of cementite with an indication of the lattice parameters. This will increase the information content of the article.
    • I have added detailed explanation of the structure of cementite with references instead of replacing the schematic illustration in Fig.9. 

Reviewer 2 Report

In this manuscript, the deformation texture of bulk cementite was investigated using scanning electron microscope, electron back scatter diffraction, nuetron diffraction and uniaxial compression. It is interestin that the (010) fiber texture was formed along compressive axis after appling compressive strain of 0.5 at 866k. However, there still exist many problems and lacking in-depth analysis. It is recommended to reconsider after major revision. The specific comments are as follows:

1)      The abbreviation of “PCS”, first appearing at the Line 61 of P2, should be given the full name.

2)      At the Line 92 of P3, the “96%” should be “96 vol.%”, and the “a few vol% of ferrite” should be“a few of ferrite” .

3)      Figure 7 and 8 showed that there are (010) tuxture along the compressive axis after deformation, which did not illustrate the primaly slip phane is the (010) plane.

Author Response

Dear reviewer,

Thank you very much for careful reading and useful comments. Our response to your comments are as follows

  • 1)      The abbreviation of “PCS”, first appearing at the Line 61 of P2, should be given the full name.

    2)      At the Line 92 of P3, the “96%” should be “96 vol.%”, and the “a few vol% of ferrite” should be“a few of ferrite” .

    • I have revised these parts after your comment. Thank you for your careful reading.
  • Figure 7 and 8 showed that there are (010) texture along the compressive axis after deformation, which did not illustrate the primary slip plane is the (010) plane.
    • In my knowledge, primary slip plane generally aligns perpendicular to the compression axis as was observed in our result. For example, after compression test of magnesium, primary slip plane 0001 aligns perpendicular to compressive axis (e.g. S.R. Agnew, Acta Mater (2001) ). Such deformation texture is observed when CRSS of primary slip plane is considerably lower comparing with other slip planes. In the case of cementite, as mentioned in the manuscript, (010) is only plane where covalent bonds do not involve. It is natural to expect that CRSS of (010) is smaller than the other slip planes. 

Reviewer 3 Report

The present manuscript studied deformation texture of bulk cementite. I recommend the paper for publication after some improvements. The paper reports an interesting and very useful work, well structured in the manuscript, but the manuscript has some weaknesses. The article needs to be expanded in practically all sections. Mentioned below aspects must be taken into consideration during the revision:

Nomenclature:

(1) I suggest adding "Nomenclature" section (with units and abbreviations) in the manuscript.

Introduction:

(2) Literature analysis should be expanded. A lot of works dealing with this issue have been published (with an emphasis on the practical side), especially in Materials journal;

(3) There must be included who did what and what he found.

Materials and Methods:

(4) An additional description of the test equipment (eg SEM) would be useful;

(5) What software was used in this project?

Results and discussion:

(6) The main limitations of the present method must be identified and discussed in the end of this section.

Conclusion:

(7) This section (Conclusion) is missing! It should be added;

(8) The conclusions should be in a quantified form.

References:

(9) References section should be extended. I propose to add a few entries in the Introduction section: regarding the microstructure and texture evolution, especially published in Materials journal (Szala et al., 2021); but also taking into account the measurement the surface topography (Macek et al., 2021; Martelo et al., 2019); and other works.

- Macek, W., Branco, R., Costa, J.D., Trembacz, J., 2021. Fracture Surface Behavior of 34CrNiMo6 High-Strength Steel Bars with Blind Holes under Bending-Torsion Fatigue. Materials 2022, Vol. 15, Page 80 15, 80. https://doi.org/10.3390/MA15010080

- Martelo, D., Sampath, D., Monici, A., Morana, R., Akid, R., 2019. Correlative analysis of digital imaging, acoustic emission, and fracture surface topography on hydrogen assisted cracking in Ni-alloy 625+. Engineering Fracture Mechanics 221, 106678. https://doi.org/10.1016/J.ENGFRACMECH.2019.106678

- Szala, M., Chocyk, D., Skic, A., Kamiński, M., Macek, W., Turek, M., 2021. Effect of Nitrogen Ion Implantation on the Cavitation Erosion Resistance and Cobalt-Based Solid Solution Phase Transformations of HIPed Stellite 6. Materials 2021, Vol. 14, Page 2324 14, 2324. https://doi.org/10.3390/MA14092324

Author Response

Dear reviewer,

Thank you very much for careful reading of the manuscript and giving useful comments. Our response to your comments are as follows.

  • I suggest adding "Nomenclature" section (with units and abbreviations) in the manuscript.

    • Thank you for your suggestion, but I decided not to add nomenclature section because this manuscript is short and not using many abbreviations. I have re-checked throughout the manuscript and revised to give full name when an abbreviation first appears.

  • Literature analysis should be expanded. A lot of works dealing with this issue have been published (with an emphasis on the practical side), especially in Materials journal. There must be included who did what and what he found.

    • This manuscript is focusing on the property of cementite which is one of common phases in steel. Regarding your suggestion, I have added five references relevant to the properties of cementite. I have searched literature of Materials journal with a keyword cementite and found 32 papers. Unfortunately, while there are many high quality papers, I could not find reference appropriate for this manuscript.

  • An additional description of the test equipment (eg SEM) would be useful. What software was used in this project?

    • I have added information of the test equipment and software.

  • The main limitations of the present method must be identified and discussed in the end of this section.   
    • I have added some explanation about limitation of present method.
  • This section (Conclusion) is missing! It should be added. The conclusions should be in a quantified form.

    • I have added conclusion section. 
  • References
    • Thank you very much for introducing interesting papers. I believe that suggested papers are very important in the field of materials science. With all due respect, however, I cannot find something in common with my manuscript from suggested papers; these papers are focusing on erosion resistance of Co-based alloy, bending-torsion fatigue of high strength steel, and hydrogen-assisted cracking in Ni-based alloy. Therefore, I could not refer these introduced papers. As I mentioned earlier, I have added references to expand introduction section. I hope revised manuscript is acceptable.

Round 2

Reviewer 3 Report

The authors attempted to provide a revision manuscript according to the reviewers' comments. They also responded to all cases individually. Although not all of their answers were satisfactory, they are generally acceptable in the scoring.